# Is Whole-Body Cryostimulation an Effective Add-On Treatment in Individuals with Fibromyalgia and Obesity? A Randomized Controlled Clinical Trial

**DOI:** 10.3390/jcm11154324

**Published:** 2022-07-26

**Authors:** Giorgia Varallo, Paolo Piterà, Jacopo Maria Fontana, Michele Gobbi, Marco Arreghini, Emanuele Maria Giusti, Christian Franceschini, Giuseppe Plazzi, Gianluca Castelnuovo, Paolo Capodaglio

**Affiliations:** 1Department of Medicine and Surgery, University of Parma, 43121 Parma, Italy; giorgia.varallo@unipr.it (G.V.); christian.franceschini@unipr.it (C.F.); 2Orthopedic Rehabilitation Unit and Research Laboratory in Biomechanics, Rehabilitation and Ergonomics, Istituto Auxologico Italiano, IRCCS, San Giuseppe Hospital, 28824 Piancavallo, Italy; p.pitera@auxologico.it (P.P.); m.gobbi@auxologico.it (M.G.); m.arreghini@auxologico.it (M.A.); p.capodaglio@auxologico.it (P.C.); 3International Center for the Assessment of Nutritional Status (ICANS), Department of Food, Environmental and Nutritional Sciences (DeFENS), University of Milan, 20133 Milan, Italy; 4Psychology Research Laboratory, Istituto Auxologico Italiano IRCCS, Ospedale San Luca, 20149 Milan, Italy; emanuelemaria.giusti@unicatt.it; 5IRCCS Istituto Delle Scienze Neurologiche di Bologna (ISNB), 40139 Bologna, Italy; giuseppe.plazzi@isnb.it; 6Department of Biomedical, Metabolic and Neural Sciences, University of Modena and Reggio Emilia, 41125 Modena, Italy; 7Department of Psychology, Catholic University of Milan, 20123 Milan, Italy; gianluca.castelnuovo@unicatt.it; 8Psychology Research Laboratory, Istituto Auxologico Italiano IRCCS, San Giuseppe Hospital, 28824 Piancavallo, Italy; 9Department of Surgical Sciences, Physical Medicine and Rehabilitation, University of Torino, 10121 Torino, Italy

**Keywords:** fibromyalgia, chronic pain, pain, sleep, depression, obesity, cryostimulation, rehabilitation

## Abstract

Pain severity, depression, and sleep disturbances are key targets for FM rehabilitation. Recent evidence suggests that whole-body cryostimulation (WBC) might be an effective add-on treatment in the management of FM. The purpose of this study was to evaluate the effects of an add-on WBC intervention to a multidisciplinary rehabilitation program on pain intensity, depressive symptoms, disease impact, sleep quality, and performance-based physical functioning in a sample of FM patients with obesity. We performed a randomized controlled trial with 43 patients with FM and obesity undergoing a multidisciplinary rehabilitation program with and without the addition of ten 2-min WBC sessions at −110 °C over two weeks. According to our results, the implementation of ten sessions of WBC over two weeks produced additional benefits. Indeed, both groups reported positive changes after the rehabilitation; however, the group that underwent WBC intervention had greater improvements in the severity of pain, depressive symptoms, disease impact, and quality of sleep. On the contrary, with respect to performance-based physical functioning, we found no significant between-group differences. Our findings suggest that WBC could be a promising add-on treatment to improve key aspects of FM, such as pain, depressive symptoms, disease impact and poor sleep quality.

## 1. Introduction

Fibromyalgia (FM) is a chronic and debilitating syndrome characterized by widespread pain, sleep disturbances, severe fatigue, depression, and cognitive impairment [1,2]. FM leads to reduced health-related quality of life [3], as well as to a disruption of recreational, social, and professional activities [4,5] with significant personal and societal costs [3]. Evidence suggests that FM affects 1.78% of the general population [6,7], mainly women, with a ratio of about 3:1 [8,9,10]. Although the pathophysiology is unknown, emerging research suggests that a combination of biological, psychological [11,12,13], and social factors [14] leads to pain amplification, disability, and central sensitization to peripheral stimuli [15], in line with the biopsychosocial model of pain [16,17].

FM, on the other hand, encompasses more than just pain, and depression and sleep disturbances should be prioritized for FM rehabilitation [18,19]. Depressive symptomatology is frequently reported by patients with FM [20,21], and it might be considered a result of chronic pain and its related limitations [22]. However, it has been suggested that depression and FM share common pathophysiological mechanisms, which could account for their high co-occurrence [23]. Importantly, patients with both FM and depressive symptoms have reduced physical functioning [24], lower quality of life [25], and poorer sleep quality [25], indicating that depression may exacerbate the effects of FM-related symptoms.

FM patients report more sleep disturbances compared to the general population and individuals with rheumatic disorders [26,27]. The association between sleep and chronic pain is well established [28]; indeed, they have a mutual influence: pain can lead to sleep difficulties, and sleep disturbances can, in turn, contribute to increased perceived pain [29,30]. Interestingly, sleep disturbances in FM are also associated with depressive symptoms [19] and patients with both FM and depression reported more sleep problems [31]. These findings suggest that pain, depression, and sleep are intertwined and might generate a vicious cycle that impacts the quality of life and functioning.

This vicious cycle is especially problematic in FM patients with comorbid obesity. Obesity affects approximately 40% of individuals with FM [32], and these two conditions interact negatively. Indeed, when comorbid obesity is present, mood and sleep disturbances and functional impairments are further exacerbated. Individuals presenting with both obesity and FM experience higher pain, depressive symptomatology, lower physical functioning [33,34,35], and poorer sleep quality [34].

Given the wide range of symptoms and consequences of FM, the management of this condition requires a multidisciplinary approach that includes pharmacotherapy, physical therapy, and cognitive behavioral therapy [36]. Non-pharmacologic interventions (e.g., exercise [37], cognitive behavioral therapy [38], mindfulness [37]) have attracted considerable interest since they may help reduce the burden of medication while improving symptomatology and quality of life [39]. By broadening the scope of those interventions, clinicians would be able to develop personalized treatment plans by selecting options also based on patients’ needs and preferences, ultimately favoring adherence to prescriptions.

Whole Body Cryostimulation (WBC) may be a useful treatment for addressing the nodes of this vicious cycle between pain, mood, and sleep [40,41,42]. WBC consists of brief (2–3 min) and repeated exposure to extremely low temperatures in specially designed cryochambers with temperatures ranging from −110 to −160 degrees Celsius. The sudden stimulation of dermal thermoreceptors induces a chain reaction of events ranging from vasoconstriction [43] to slowed nerve conduction velocity of slow conducting C fibers, disabling the sensory receptors as well as their connections with the proprioceptors [44] to pain modulation via inhibitory action in conjunction with the cold-induced release of endorphins [45] and norepinephrine producing an overall analgesic effect. Repeated exposure to WBC seems to reduce the production of pro-inflammatory and oxidative markers, whereas the anti-inflammatory and anti-oxidative compounds are produced in larger quantities [46,47]. For example, WBC has been shown to influence cytokine levels, specifically increasing IL-6 and IL-10 while decreasing IL-1 [48,49]. All of these mechanisms may account for the reported pain-relieving properties of WBC. Thus, WBC could be a particularly useful treatment for FM as an imbalance of pro- and anti-inflammatory cytokines is thought to play a role in the development and maintenance of pain as well as the occurrence of many of the disease’s clinical features [50], especially when comorbid obesity is present, which is associated with chronic systemic inflammation [51,52].

To date, most studies have been focused on the effect of WBC on FM, reducing pain severity [40,53], impact of the disease [40], and improving quality of life [40,53,54]. However, preliminary evidence suggests that WBC may also improve sleep quality [42,54], physical functioning [55], and reduce depressive symptoms [41]. These findings suggest that WBC may be especially beneficial for patients with obesity and FM, who have higher levels of pain severity, depressive symptoms, impact of disease, and poorer sleep quality [50,56,57]. To date, however, no study has yet been conducted to evaluate the effect of WBC on these aspects in patients with FM and comorbid obesity.

Thus, this study aimed to determine whether the addition of WBC to a multidisciplinary rehabilitation program could result in a greater reduction in pain intensity, depressive symptoms, disease impact, and improvement in sleep quality and performance-based physical functioning in a sample of FM patients with obesity. We hypothesized that participants receiving an add-on WBC intervention would show greater improvements in (i) pain severity, (ii) disease impact, (iii) depressive symptomatology, (iv) sleep quality, and (v) performance-based physical functioning as compared to the controls.

## 2. Materials and Methods

We conducted a randomized controlled trial on patients with FM and obesity who underwent a multidisciplinary rehabilitation program with or without the addition of WBC. Eligible participants were recruited consecutively from the in-patients of the Rehabilitation Unit of the IRCCS Istituto Auxologico Italiano (Piancavallo, Italy). The hospitalization aimed at weight management and physical conditioning. The stages of the protocol are described in Figure 1.

Data were collected from 1 June 2021 to 30 December 2021. Anthropometric values, body composition, demographic, and clinical data were collected at admission and at discharge within 4 weeks. Primary outcomes were assessed before the experiment (PRE-T0) and two weeks later (POST-T1) at the end of the study protocol. Subjects assigned to RWBC were familiarized with WBC at PRE-T0. Then, they underwent 10 sessions of WBC over a 2-week period (1 treatment per day, Monday through Friday, at 8:15 a.m., before exercise classes and physical therapy). The multidisciplinary rehabilitation program consisted of a 3-h daily program including sessions of physiotherapy, nutritional and psychological support, and adapted physical activity classes (Figure 2).

The following inclusion criteria were used: (i) age between 18 and 65 years; (ii) FM diagnosed by a rheumatologist according to the criteria of the American College of Rheumatology criteria; (iii) FM diagnosed for more than one year; meeting the Fibromyalgia research criteria measure with the Fibromyalgia Survey Questionnaire, Italian version [58]. Participants were excluded if they had: (i) severe psychiatric conditions, (ii) acute respiratory diseases, acute cardiovascular diseases, unstable hypertension, cold intolerance, claustrophobia, or pregnancy, (iii) recent modification of the usual pharmacological treatment, or (iv) previous WBC and (v) body temperature greater than 37.5 °C. Participants were then randomly assigned to either a “rehabilitation + WBC” (RWBC) or “rehabilitation” intervention (R) using a randomization scheme generated through the http://www.randomization.com website (accessed on 29 May 2021). Concealed allocations were performed by an independent researcher to prevent selection bias. This study was an open trial; thus, both participants and researchers knew which treatment the patient is receiving (i.e., WBC). Each participant received verbal and written information about the research protocol. All participants signed an informed consent form in which the study procedures were explained in accordance with the Declaration of Helsinki. The Ethics Committee of the Istituto Auxologico Italiano approved the study protocol and materials (code 2021_05_18_14).

### 2.1. Procedures Description

Rehabilitation program. The multidisciplinary rehabilitation program consisted of individual nutritional intervention, psychological support, and supervised physical activity throughout hospitalization. All patients received a balanced hypocaloric diet consisting of 18–20% proteins, 27–30% fats (<8% saturated fat), 50–55% carbohydrates (<15% simple sugars), and 30 g of fibers from fresh vegetables. The diet plan was organized by the hospital dietitian into three meals (breakfast, lunch, and dinner) with an energy distribution of 20%, 40%, and 40%, respectively. Two daily 60-min physiotherapy sessions were performed under the supervision of a physiotherapist and consisted of: personalized progressive aerobic training, postural control exercises, and progressive strengthening exercises.

WBC intervention. Patients assigned to the RWBC group were exposed to extremely cold, dry air at −110 °C for 2 min in a cryochamber (Artic, CryoScience, Rome, Italy) located at the Rehabilitation Unit of San Giuseppe Hospital, Istituto Auxologico Italiano, in Piancavallo, Italy. Patients minimally dressed, wearing a surgical mask, earband, gloves, t-shirt, shorts, socks, and plastic clogs, were first familiarized with the cryochamber (Artic, CryoScience, Rome, Italy) with a first 1-min session at −110 °C. Before entering the chamber, eventual glasses, contact lenses, and metallic jewelry were removed, and the body thoroughly dried. Regular vocal and eye contact with the patient was maintained during the session. The following sessions lasted 2 min at a temperature of −110°. The patient’s skin surface temperature was measured before and after each treatment with an infrared thermometer (Fluke Corporation, Everett, WA, USA) at the neck, quadriceps, popliteal fossa, and calf level.

### 2.2. Measures

#### 2.2.1. Anthropometrics, Demographic and Clinical Characteristics

Baseline demographic and clinical characteristics, collected at admission (PRE-T0) and discharge (POST-T1), included age, gender, body mass index (BMI; calculated as kg/m^2^ where kg is a person’s weight in kilograms and m^2^ is their height in meters squared), and pain duration.

#### 2.2.2. Primary Outcomes Measures

The following outcome measures were collected at baseline (T0) and after two weeks (T1).

Pain severity. Perceived pain was assessed using a Pain Numeric Rating Scale (PNRS) ranging from 0 to 10, where high scores indicate high levels of pain intensity [56].

Disease impact. The Fibromyalgia Impact Questionnaire (FIQ) [57], in its Italian validation [59], was used to measure the overall impact of the disease. FIQ is a self-administered test that measures physical function, work status, depression, anxiety, sleep, pain, stiffness, fatigue, and well-being. Total scores range from 0 to 100, where higher scores indicate a more severe impact of FM.

Depressive symptomatology. The Beck Depression Inventory (BDI-II) [60] is a self-reported questionnaire used for the evaluation of the severity of depressive symptoms. The BDI-II consists of 21 items scored between 0 (lack of symptoms) and 3 points (the highest severity of the described symptom). Total scores between 0 and 11 indicate a lack of depressive symptoms, 12–26 indicate a mild depressive episode, 27–49 indicate a moderate depressive episode, and 50–63 indicate a severe depressive episode. We used the Italian validation of this measure [61] which has good psychometric properties in line with the original version.

Sleep quality. To assess perceived sleep quality, we used a Visual Analogue Scale from 0 to 10, where 0 corresponds to “worst possible sleep quality” and 10 “excellent sleep quality”.

Performance-based physical functioning. The six-minute walk test (6MWT) was performed indoors, along a long, flat, undisturbed 30-m hospital corridor with the length marked every 5 m according to the “ATS Statement: Guidelines for the Six-Minute Walk Test”. Chest pain, severe dyspnea, physical exhaustion, muscle cramps, sudden gait instability, or other signs of severe distress were additional criteria for stopping the test. A higher score corresponds to greater walking endurance [62]. In addition, the timed up and go (TUG) test was used to evaluate balance and functional mobility. The participant is asked to stand up from a standard chair, walk 3 m, turn around, return to the chair, and sit down. The score corresponds to the time (seconds) to perform the test [63]. Given the patients’ condition of severe obesity, the 6MWT and TUG were optional and were only carried out with the patients’ consent.

### 2.3. Statistical Analysis

A sample size of 34 participants was deemed sufficient to detect a significant interaction between groups by time on outcomes, with an effect size of 0.25, a power of 80%, and an alpha level of 0.05.

Summary statistics were used to describe the sample (mean and standard deviation, frequencies, and percentages). Shapiro–Wilk tests, Q-Q plots, and histograms were used to evaluate the normality of the distribution of the PNRS, FIQ, BDI, VAS-Sleep, 6MTW, and TUG. Because the assumption of normality of the data was not respected, we decided to use a nonparametric approach. Baseline differences between the RWBC group and the R group in age, BMI, and pain duration were inspected using Mann–Whitney U tests. Then, a nonparametric ANOVA-type test was employed to assess the interaction between time and group on the dependent variables (i.e., NPRS, FIQ, BDI, VAS sleep quality, 6MWT, and TUG). The analyses of the time x group interactions were performed using the R (version 4.0.1) package nparLD [64]. α-levels were set at 0.05 for all analyses.

## 3. Results

### 3.1. Participants’ Characteristics at Baseline

Table 1 shows the baseline clinical characteristics of the participants. We enrolled 20 female patients in the RWBC (mean age: 52.82 ± 7.78) and 23 in the R (mean age: 48.7 ± 7.15). There were no significant differences between the two groups at baseline in terms of age, BMI, duration of pain, and scores on questionnaires and performance-based tests (i.e., PNRS, FIQ, BDI, VAS sleep quality, 6MWT, and TUG). Participants were closely monitored, and no episodes of treatment-emergent adverse events were observed. Every day before entering the cryochamber, participants were examined by a physician. All the participants completed the whole study. All patients in the RWBC group underwent all 10 sessions of cryostimulation. In the RWBC group, there was no missing data. In contrast, in the R group, 13 subjects did not consent to perform the 6MWT, and 7 subjects did not consent to undergo the TUG test. Analyses of this outcome were therefore conducted on 16 subjects.

### 3.2. Results of Nonparametric ANOVA-Type Test

The results of the nonparametric ANOVA-type test are presented in the following paragraphs. For each outcome, both the main effect of time and the interaction effect between group and time. The median and interquartile ranges (IQR) of the outcomes of the RWBC and R group at T0 and T1 are reported in Table 2.

### 3.3. Pain Severity

There was no difference in the scores on PNRS between groups at T0, U = 305, *p* = 0.62. The main effect of time showed a statistically significant difference at the different time points in PNRS, Anova-Type Statistics (ATS) = 278.92, *p* < 0.0001, and also a statistically significant interaction between the group and time on PNRS, ATS = 56.42, *p* < 0.001, indicating that the RWBC group reported a significantly larger decrease in pain severity levels compared with the R group (see Figure 3).

### 3.4. Impact of Disease

At baseline, no difference between groups in the scores on FIQ was found, U = 253, *p* = 0.547. The main effect of time revealed a statistically significant difference at T1 compared to T0 on FIQ, ATS = 62.35, *p* < 0.001, suggesting that both groups at T1 reported an improvement in the impact of illness. However, an interaction effect group and time was found, ATS = 14.23, *p* < 0.001, indicating that the RWBC group had a greater reduction in the impact of illness compared to the R group (see Figure 4).

### 3.5. Depressive Symptomatology

According to our results, no difference in BDI scores was found at baseline between the R and RWBC groups, U = 248, *p* = 0.651. The main effect of time was significant, suggesting that both groups experienced a reduction in depressive symptomatology, ATS = 46.03, *p* < 0.001. Nevertheless, also in this case, the interaction group by time was significant, and the RWBC group reported a greater decrease in depressive symptomatology after WBC intervention compared to the R group, ATS = 21.38, *p* < 0.001 (see Figure 5).

### 3.6. Sleep Quality

Finally, regarding sleep quality, there was no difference at baseline between the two groups, U = 183, *p* = 0.247. The significant main effect of time, ATS = 78.44, *p* < 0.001, suggests that both groups at T1 reported higher levels of sleep quality compared to T0. Importantly, there was a statistically significant interaction between the group and time, ATS = 29.85, *p* < 0.001, indicating that sleep quality was significantly higher in the RWBC group at T1 compared to the R group (see Figure 6).

### 3.7. Performance-Based Physical Functioning

At baseline no significant difference was found for both 6MWT (U = 101, *p* = 0.725) and TUG (U = 101, *p* = 0.710) between group. There was no statistically significant interaction between the intervention and time on 6MWT, ATS = 0.11, *p* = 0.741. However, the main effect of time showed a statistically significant difference in median 6MWT at T1 compared to T0, ATS = 76.98, *p* < 0.0001 (see Figure 7). Additionally, there was no statistically significant interaction between the group and time on TUG, ATS = 1.20, *p* = 0.27. Even in this case, there was a statistically significant main effect of time ATS = 76.98, *p* < 0.001 (see Figure 8).

## 4. Discussion

This study aimed to evaluate the effects of WBC on pain severity, impact of illness, depressive symptomatology, sleep quality, and performance-based physical functioning (i.e., 6MWT and TUG) in patients with FM and obesity. We hypothesized that combining WBC with standard treatment would more effectively reduce pain severity, depressive symptomatology levels, and overall impact of the disease and improve sleep quality and performance-based physical functioning as compared to standard treatment alone. Our hypotheses were partially confirmed by the results. The implementation of ten WBC sessions over two weeks resulted in additional benefits. Indeed, both groups reported positive changes in all outcomes assessed after the rehabilitation period, but the magnitude of changes was statistically greater in the RWBC group, particularly for pain severity, depressive symptomatology, illness impact, and sleep quality. In performance-based physical functioning, however, we found no differences between the two interventions, with both groups improving at the end of the intervention but no differences between them. Nevertheless, our preliminary findings suggest that WBC may be a useful adjunct treatment for simultaneously improving several key aspects of FM and obesity.

Our results are partially consistent with previous evidence. Rivera et al. [40] found a significant reduction in pain severity, and impact of the disease in 60 patients with FM after 10 WBC sessions. This study used a crossover design in which patients undergoing WBC were used as controls after a wash-out period. However, the authors pointed out the limitations of this method, suggesting that the wash-out period was insufficient to ensure that the effect of the treatment disappeared. On the contrary, in our study, a control group was used, allowing for more reliable conclusions.

A recent prospective randomized, double-blind study found that 10 sessions of WBC significantly reduced depressive symptoms in adult patients who suffered a depressive episode [41]. In contrast, in this study, the authors did not report any improvement in sleep quality [41]. These differences could be attributed to the samples’ different clinical and demographic characteristics, as well as the different instruments used to assess the outcomes. However, our results on improved sleep quality after WBC are in line with other studies on competitive athletes [42,65]. Our study offers the first investigation on the effect of WBC on FM and obesity, suggesting that WBC may be beneficial in this population as well for reducing pain severity and depressive symptomatology and improving sleep quality. Including cryotherapy in the rehabilitation of patients with FM and obesity may benefit the latter condition as well. Reduced pain intensity and improved physical functioning may promote increased physical activity compliance, with potential benefits for both weight loss and pain management.

One interesting aspect of our study is that sleep, mood, and pain have all improved, in line with previous evidence suggesting that these three factors interact and influence each other [66,67,68,69]. Our research design, on the other hand, prevents us from determining the primary factor that influences others. Future research is required to disentangle the interrelationships and determine which of the factors mediates the change, potentially revealing a useful primary intervention target. This is still an open debate with contradictory evidence [30,70].

Previous research has shown that WBC can improve muscle strength, cardiovascular and hormonal systems, lipid profile, and muscle recovery after exercise resulting in an overall improvement in functional status in athletes [71]. However, no previous study has evaluated performance-based physical functioning in FM patients with comorbid obesity. Although physical functioning is a multidimensional construct with both subjective and objective aspects, most studies have relied on self-report measures. Our data show a significant improvement in physical function but fail to demonstrate whether this was due to WBC and not just to the multidisciplinary rehabilitation program. Although easy to perform, both the TUG and 6MWT may have low sensitivity in detecting changes that can be considered clinically significant. Therefore, further studies with larger sample sizes are needed to standardize the methodology and calculate test–retest reliability and minimum detectable change to better characterize the sensitivity of these tests and meet the standards needed to make decisions at the individual patient level. Another possible explanation is that two weeks of WBC is insufficient to produce a significant improvement in performance-based physical functioning. Longer WBC interventions may be evaluated in future studies.

This study has some limitations, including the lack of biomarkers documenting WBC-induced effects, the open trial design, and the pre-post design with no follow-up. Since no sham treatment was provided, a placebo effect from our WBC intervention could not be ruled out. Additionally, several participants did not consent to undergo physical tests (i.e., 6MWT, TUG), resulting in missing data for those measures. Despite these limitations, WBC appears to have the potential to enhance rehabilitation programs and their cost-effectiveness in patients with FM and obesity. Because the coexistence of FM and obesity reduces functional capacities and increases self-perceived disability, additional rehabilitative resources are required. Identifying effective add-on interventions able to boost rehabilitation programs appears to be an important issue. Not secondarily, the patients’ high compliance with the sessions and satisfaction with the treatment reported in most studies seem to make WBC a preferred component of the rehabilitation program, which appears to be critical in the long-term management of FM and obesity.

## 5. Conclusions

According to our findings, WBC is effective for reducing pain severity and improving sleep, and mood in patients with fibromyalgia and obesity. Importantly, pain reduction may facilitate compliance to physical activity intervention with a positive effect on both weight and pain management in patients with comorbid obesity. However, further studies with follow-up measures will be needed to assess the long-term beneficial effects and to explore the beneficial effect also on performance-based physical functioning.

## Figures and Tables

**Figure 1 jcm-11-04324-f001:**
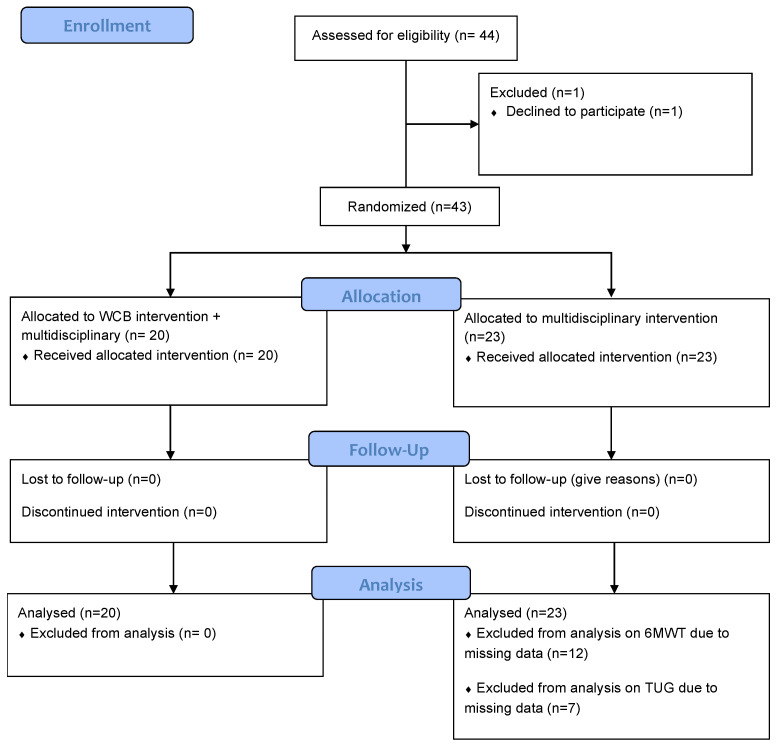
The CONSORT flow chart describing each stage of the study: enrollment, allocation, intervention exposure, follow-up, and analysis.

**Figure 2 jcm-11-04324-f002:**
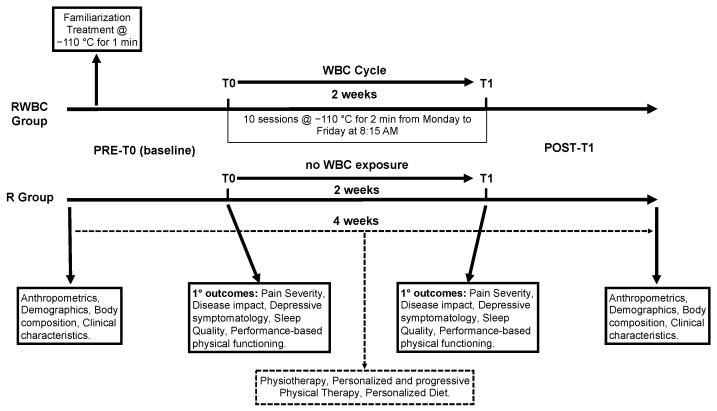
Study design with the timeline of the study protocol, the baseline measures, and the primary outcomes.

**Figure 3 jcm-11-04324-f003:**
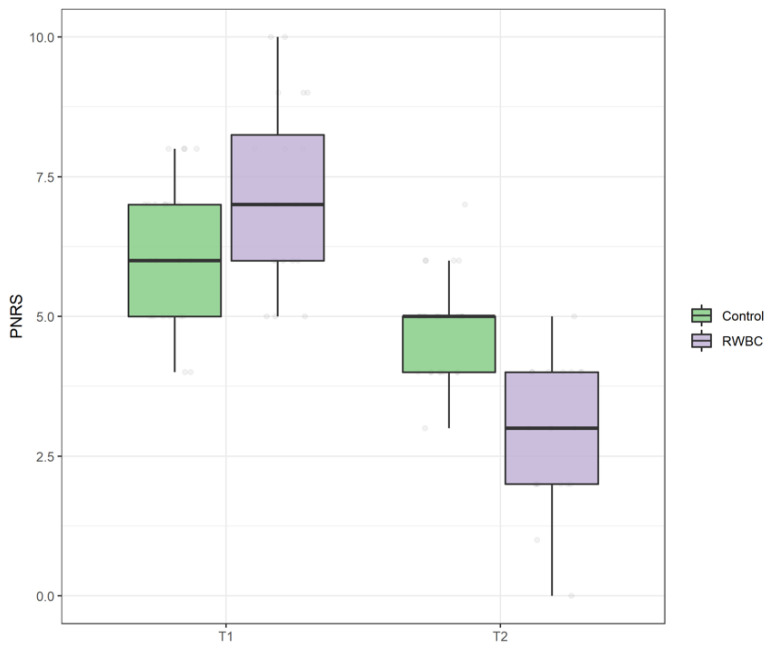
Boxplot of Pain Numeric Rating Scale (PNRS) scores at T0 and T1 in control group and intervention group (RWBC).

**Figure 4 jcm-11-04324-f004:**
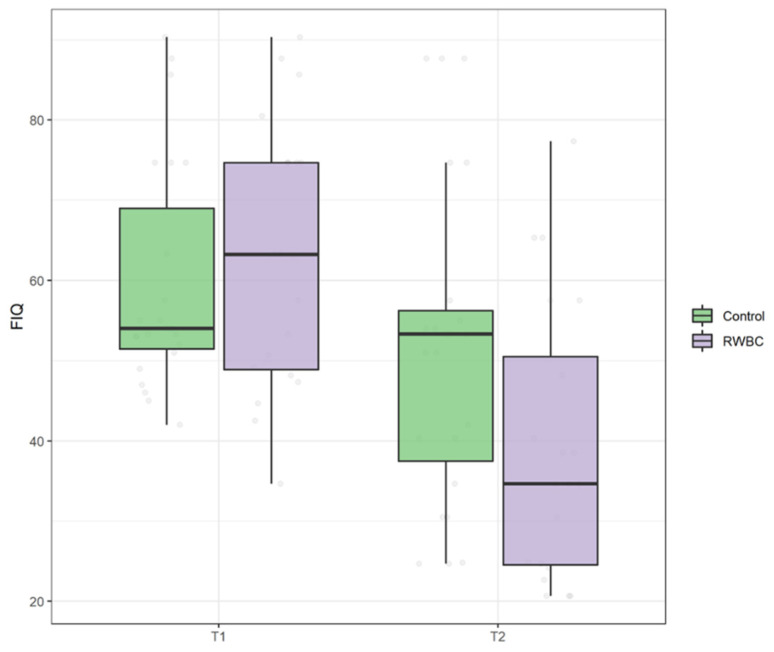
Boxplot of Fibromyalgia Impact Questionnaire (FIQ) scores at T0 and T1 in control group and intervention group (RWBC).

**Figure 5 jcm-11-04324-f005:**
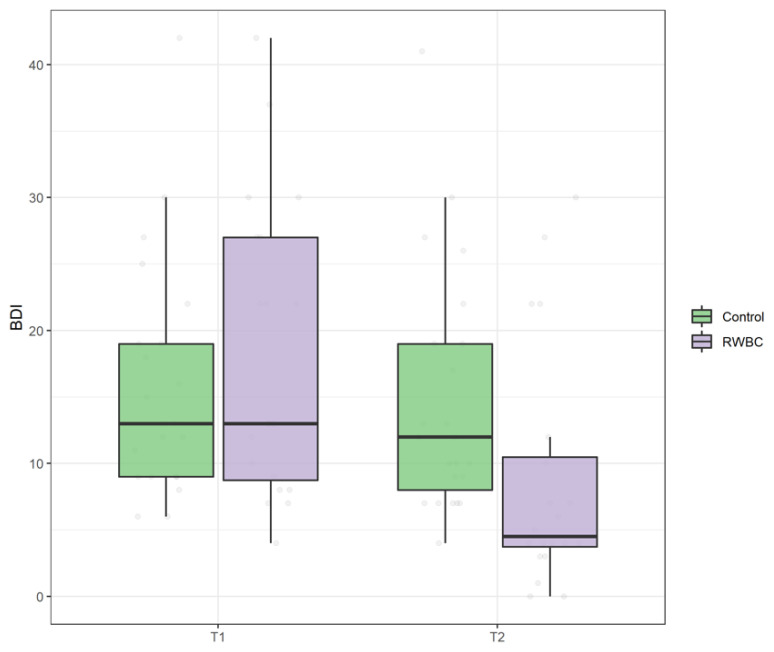
Boxplot of Beck Depression Inventory (BDI) scores at T0 and T1 in control group and intervention group (RWBC).

**Figure 6 jcm-11-04324-f006:**
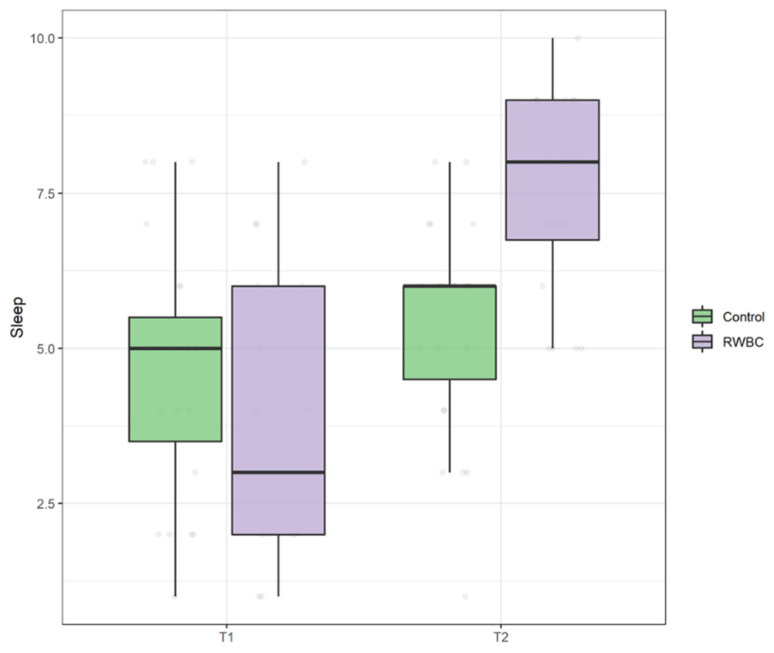
Boxplot of sleep quality scores at T0 and T1 in control group and intervention group (RWBC).

**Figure 7 jcm-11-04324-f007:**
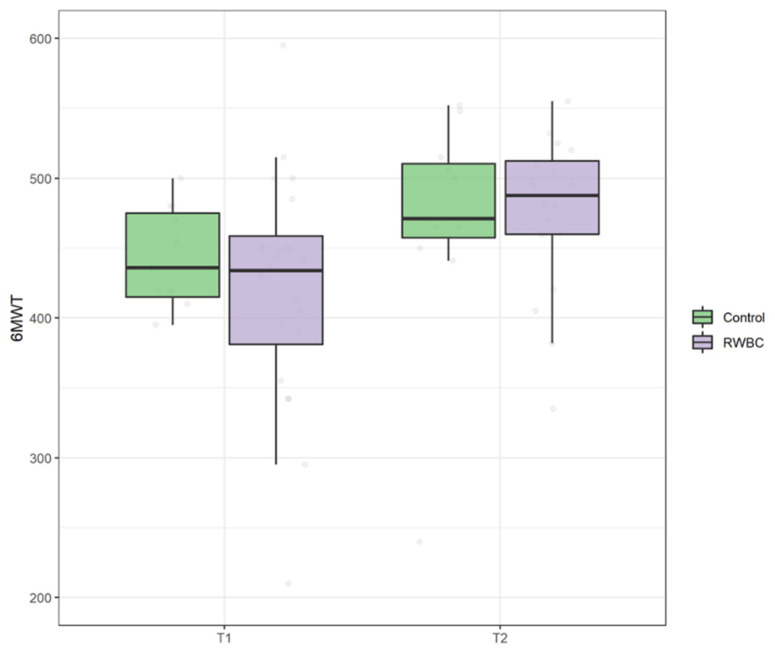
Boxplot of six-minute walking test (6MWT) scores at T0 and T1 in control group and intervention group (RWBC).

**Figure 8 jcm-11-04324-f008:**
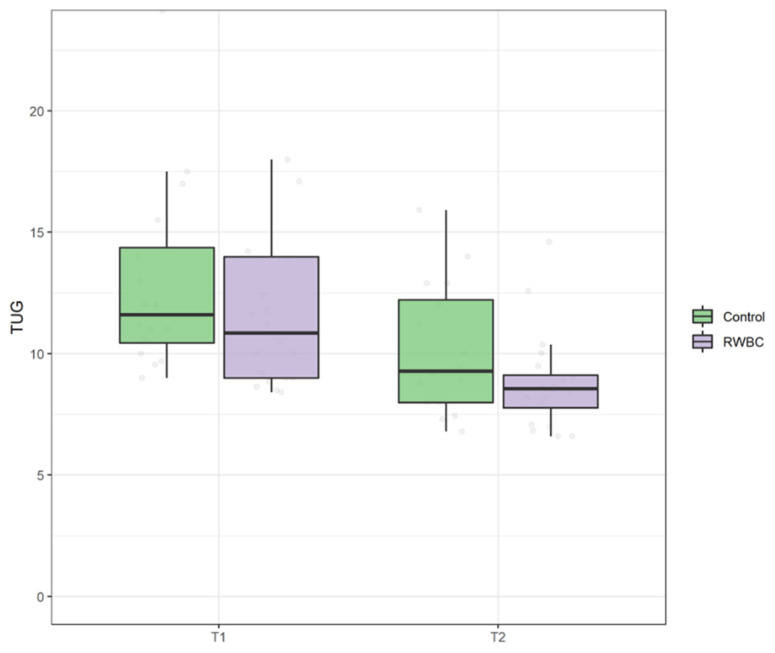
Boxplot of time up and go (TUG) scores at T0 and T1 in control group and intervention group (RWBC).

**Table 1 jcm-11-04324-t001:** Participants’ demographic and clinical characteristics at baseline.

	R Group		RWBC Group		U	*p*
	Median	IQR	Median	IQR		
Age (in years)	56	4.50	53	9.00	81.5	0.097
BMI	39.9	6.15	38	7.25	159	0.539
Pain duration (in years)	7	3.50	7	5.00	201	0.480
PNRS	6.00	2.00	7	2.25	305	0.062
FIQ	54	17.50	63.25	25.75	253	0.547
BDI	13	10.00	13	18.25	248	0.651
VAS sleep	5	2.00	3	4.00	183	0.247
6MWT	436	60	434	77.40	101	0.725
TUG	11.19	3.93	10.86	4.99	101	0.710

Note: BMI—Body Mass Index; PNRS—Pain Numeric Rating Scale; FIQ—Fibromyalgia Impact Questionnaire; BDI—Beck Depression Inventory; VAS sleep—Visual analogue scale for sleep quality; 6MWT—six-minute walking test; TUG—time up and go.

**Table 2 jcm-11-04324-t002:** Median and interquartile range (IQR) of self-reported and performance-based clinical variables at T0 and T1.

	Group	N	Median	IQR
PNRS T0	0	23	6.00	2.00
	1	20	7.00	2.25
FIQ T0	0	23	54.00	17.50
	1	20	63.25	25.75
BDI T0	0	23	13.00	10.00
	1	20	13.00	18.25
VAS sleep T0	0	23	5.00	2.00
	1	20	3.00	4.00
PNRS T1	0	23	5.00	1.00
	1	20	3.00	2.00
FIQ T1	0	23	53.33	18.75
	1	20	34.67	25.95
BDI T1	0	23	12.00	11.00
	1	20	4.50	6.75
VAS sleep T1	0	23	6.00	1.50
	1	20	8.00	2.25
6MWT T0	0	11	436.00	60.00
	1	20	434.00	77.50
TUG T0	0	16	11.19	3.93
	1	20	10.86	4.99
6MWT T1	0	11	471.00	53.00
	1	20	487.50	52.50
TUG T1	0	16	8.91	4.25
	1	20	8.82	1.35

Note: 0—R group; 1—RWBC group; T0—admission; T1—discharge; PNRS—Pain Numeric Rating Scale; FIQ—Fibromyalgia Impact Questionnaire; BDI—Beck Depression Inventory; VAS sleep—Visual analogue scale for sleep quality; 6MWT—six-minute walking test; TUG—time up and go.

## Data Availability

The data presented in this study are available on request from the corresponding author. The data are not publicly available due to privacy restrictions.

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
