# Peer review of "Is Whole-Body Cryostimulation an Effective Add-On Treatment in Individuals with Fibromyalgia and Obesity? A Randomized Controlled Clinical Trial"

_jcm, 2022, doi:10.3390/jcm11154324_

Round 1
Reviewer 1 Report
Dear authors,
The manuscript explains the effects of WBC on fibromyalgia and obesity.
In the methodology section, the authors report that this is an open trial, where everyone knows which group the participants belong to. Could this be a bias because the placebo effect exists, and half of the sample did not receive the full treatment?
Why did they choose the neck, quadriceps popliteal fossa and calf to measure temperature?
"13 subjects did not perform the 6MWT and 7 subjects did not complete the TUG", were these patients excluded from the sample? Because it is a large percentage of the group?
There are some editorial errors that should be corrected (e.g., the 21 reference in line 120).
The discussion section should be improved. The authors mention comorbidities such as obesity, but in the results and in the discussion, there is only a brief explanation.
It might also be interesting to add some explanation about physiotherapeutic treatments in fibromyalgia, even more so when conventional treatment always included hot thermotherapy and in this case the authors support the opposite. What is the reason for this change?
Author Response
We appreciate the insightful comments of Reviewer 1. We have highlighted in red the changes in the manuscript. Below every comment, you can find a point-by-point response to the reviewers' comments and concerns.
Dear authors,
The manuscript explains the effects of WBC on fibromyalgia and obesity.
Q: In the methodology section, the authors report that this is an open trial, where everyone knows which group the participants belong to. Could this be a bias because the placebo effect exists, and half of the sample did not receive the full treatment?
R: The authors thank reviewer 1 for his comment. Since no sham treatment was performed, we can’t completely rule out a possible placebo effect on the group receiving WBC. Indeed, we mention this aspect in the limitations section (see line 386).
Q: Why did they choose the neck, quadriceps popliteal fossa and calf to measure temperature?
R: The authors thank reviewer 1 for his comment. It is a standardized procedure we use to measure the reductions in muscle temperature (calf and quadriceps), lower joint temperature (popliteal fossa) and upper body temperature (neck) to roughly verify the total body cooling effect of a treatment.
Q: "13 subjects did not perform the 6MWT and 7 subjects did not complete the TUG", were these patients excluded from the sample? Because it is a large percentage of the group?
R: Since patients had moderate and severe obesity, TUG and 6MWT were made optional and performed only on patients who consented. We explained this aspect in the Methods section (see line 218), results (see line 252) and added this additional limitation in the discussion (see line 388). Indeed, we agree with the reviewer that it is a large percentage of the group, but we thought it was important to analyze the possible effect of WBC on performance-based physical functioning.
Q: There are some editorial errors that should be corrected (e.g., the 21 reference in line 120).
R: Thanks for noticing the inaccuracy. We fixed editorial errors.
Q: The discussion section should be improved. The authors mention comorbidities such as obesity, but in the results and in the discussion, there is only a brief explanation.
R: Thank you for the opportunity to further clarify this aspect. We improved the discussion better explaining the implications of WBC in patients with FM and comorbidities such as obesity.
Q: It might also be interesting to add some explanation about physiotherapeutic treatments in fibromyalgia, even more so when conventional treatment always included hot thermotherapy and in this case the authors support the opposite. What is the reason for this change?
R: We agree with reviewer 1 about the effectiveness of hot thermotherapy in relieving painful symptoms of fibromyalgia. However, because mood and sleep are also altered in fibromyalgia, with a possible further exacerbating effect of the obesity condition, cryotherapy represents an alternative that allows for intervention in multiple aspects simultaneously. Compared to hot thermotherapy, there is no evidence of improvement in mood and sleep. Adding a section on heat therapy could be confusing and is beyond the scope of our article. With respect to physiotherapeutic treatments, we have added a paragraph emphasizing the importance of the multidisciplinary approach and citing a body of work evaluating the effectiveness of physiotherapy treatments.

Reviewer 2 Report
Dear authors,
The study investigates a topic that is rarely addressed in patients with FM, presenting a new possibility for treating the main symptoms. I believe that the topic is relevant and is of interest to professionals working with FM.
Despite the strengths of the study, I make suggestions on questions that may be more accurate.
Title – Indicate in the title that this is a controlled and randomized clinical trial, according to the CONSORT check-list.
Abstract – The summary needs to be improved.
I suggest a better description of the method regarding the intervention (intervention time, quantity) and the results (presenting values of some results).
Introduction
In the introduction, the treatment alternatives can be briefly placed for contextualization. The authors say that multidisciplinary treatment is recommended and they use this approach in research, they could say so, citing some studies on these treatments (exercise, medication, CBT).
Methods
Title of figure 1, only flow chart is incomplete.
What does the control treatment consist of? A better description of what has been accomplished in physical therapy and exercise is needed.
When was the second collection performed? Immediately after intervention sessions? Hours later? Days later? This information is essential, as it directly implies the interpretation of the results.
I believe it is essential to follow the CONSORT.
Results
In the title of table 1 there is a description in addition to the title, I suggest putting the description of the result in the previous paragraph.
Discussion
As the study is with patients with obesity, this issue should be better addressed in the discussion. Based on the results found and discussed, cryotherapy would be efficient regardless of whether the patients were obese or not. Thus, given the numerous difficulties that obese patients face in treatment (with exercise and nutrition, for example), the use of a passive intervention (cryotherapy) can have a great impact on clinical practice.
In addition, or complementing my previous comment, I suggest putting a paragraph talking about the practical implications of these results. Since results with cryotherapy alone are fragile.
Specific comments
Line 52: replace fibromyalgia with FM.
Line 114: use the term (T1) and (T2), however in line 156 they use (T0) and (T1), check.
Lines 303 and 304: “It is possible that the differences in the effect on sleep quality are due to the samples' different clinical 304 and demographic characteristics”, in addition, the instruments used were different, also justifying the data.
Lines 336 e 337: “Despite those limitations, initial evidence in the literature indicates that WBC effectively reduces FM symptoms” – WBC together with other treatments, the isolated effect was not verified.
Lines 343-345: ‘Not secondarily, the patients' high compliance to the sessions and satisfaction about the treatment reported in most studies seem to make WBC a preferred component of the rehabilitation program’, - you need to reference. “..which appears crucial in the long-term management of FM and obesity.”
Author Response
We appreciate the insightful comments of Reviewer 2. We have highlighted in red the changes in the manuscript. Below every comment, you can find a point-by-point response to the reviewers' comments and concerns.
Dear authors,
The study investigates a topic that is rarely addressed in patients with FM, presenting a new possibility for treating the main symptoms. I believe that the topic is relevant and is of interest to professionals working with FM.
Despite the strengths of the study, I make suggestions on questions that may be more accurate.
Q: Title – Indicate in the title that this is a controlled and randomized clinical trial, according to the CONSORT check-list.
R: We thank Reviewer 2 for the suggestion. We changed the title accordingly.
Q: Abstract – The summary needs to be improved.
I suggest a better description of the method regarding the intervention (intervention time, quantity) and the results (presenting values of some results).
R: We thank Reviewer 2 for the comment. The abstract summary has been improved as suggested. The intervention time, amount, and summary of results (in terms of improvement or not) have been introduced. However, we decided not to report the values of the results, as they would have taken up too much space, given the multiple outcomes, exceeding the 200-word limit.
Q: Introduction
In the introduction, the treatment alternatives can be briefly placed for contextualization. The authors say that multidisciplinary treatment is recommended and they use this approach in research, they could say so, citing some studies on these treatments (exercise, medication, CBT).
R: We thank Reviewer 2 for the valuable comment. As suggested, we have introduced a short paragraph describing multidisciplinary interventions used nowadays (see line 79).
Q: Methods
Title of Figure 1, only flow chart is incomplete.
R: We improved the title that describes the purpose of the flow chart.
Q: What does the control treatment consist of? A better description of what has been accomplished in physical therapy and exercise is needed.
R: We thank Reviewer 2 for the valuable suggestion. We have introduced a paragraph that better describes the intervention.
Q: When was the second collection performed? Immediately after intervention sessions? Hours later? Days later? This information is essential, as it directly implies the interpretation of the results.
I believe it is essential to follow the CONSORT.
R: We followed the CONSORT guidelines and we clarified all the issues suggested by the reviewer by introducing a schematic representation of the study design showing the timeline of the study (see Figure 2).
Q: Results
In the title of table 1 there is a description in addition to the title, I suggest putting the description of the result in the previous paragraph.
R: Thank you for the suggestion. We modified the text accordingly.
Q: Discussion
As the study is with patients with obesity, this issue should be better addressed in the discussion. Based on the results found and discussed, cryotherapy would be efficient regardless of whether the patients were obese or not. Thus, given the numerous difficulties that obese patients face in treatment (with exercise and nutrition, for example), the use of a passive intervention (cryotherapy) can have a great impact on clinical practice.
In addition, or complementing my previous comment, I suggest putting a paragraph talking about the practical implications of these results. Since results with cryotherapy alone are fragile.
R: We followed the reviewer's suggestion and implemented the conclusion with a few practical implications of these results (see line 355 and 397).
Specific comments
Q: Line 52: replace fibromyalgia with FM.
R: We replaced it.
Q: Line 114: use the term (T1) and (T2), however in line 156 they use (T0) and (T1), check.
R: We kept the terms (T0) and (T1)
Q: Lines 303 and 304: “It is possible that the differences in the effect on sleep quality are due to the samples' different clinical and demographic characteristics”, in addition, the instruments used were different, also justifying the data.
R: We pointed out that these differences could be caused by different instruments used to assess outcomes.
Q: Lines 336 e 337: “Despite those limitations, initial evidence in the literature indicates that WBC effectively reduces FM symptoms” – WBC together with other treatments, the isolated effect was not verified.
R: We agree with the reviewer and have modified the sentence accordingly.
Q: Lines 343-345: ‘Not secondarily, the patients' high compliance to the sessions and satisfaction about the treatment reported in most studies seem to make WBC a preferred component of the rehabilitation program’, - you need to reference. “..which appears crucial in the long-term management of FM and obesity.”
R: We referenced the review article published by Fontana et al 2022.

Round 2
Reviewer 1 Report
Dear authors,
Thank you for the effort to make the proposed changes.
Methodology section, results and conclusions have been improved.
Figure 2 aids the understanding of the procedure performed.
Congratulations.